# Comparative Interaction Studies of Quercetin with 2-Hydroxyl-propyl-β-cyclodextrin and 2,6-Methylated-β-cyclodextrin

**DOI:** 10.3390/molecules27175490

**Published:** 2022-08-26

**Authors:** Vasiliki Vakali, Michail Papadourakis, Nikitas Georgiou, Nikoletta Zoupanou, Dimitrios A. Diamantis, Uroš Javornik, Paraskevi Papakyriakopoulou, Janez Plavec, Georgia Valsami, Andreas G. Tzakos, Demeter Tzeli, Zoe Cournia, Thomas Mauromoustakos

**Affiliations:** 1Organic Chemistry Laboratory, Department of Chemistry, National and Kapodistrian University of Athens, Panepistimiopollis Zografou, 11571 Athens, Greece; 2Biomedical Research Foundation Academy of Athens, 4 Soranou Ephessiou, 11527 Athens, Greece; 3Department of Chemistry, Section of Organic Chemistry and Biochemistry, University of Ioannina, 45110 Ioannina, Greece; 4Slovenian NMR Centre, National Institute of Chemistry, SI-1001 Ljubljana, Slovenia; 5Department of Pharmacy, School of Health Sciences, National and Kapodistrian University of Athens, 15784 Athens, Greece; 6Institute of Materials Science and Computing, University Research Center of Ioannina (URCI), 45110 Ioannina, Greece; 7Laboratory of Physical Chemistry, Department of Chemistry, National and Kapodistrian University of Athens, Panepistimioupolis Zografou, 11571 Athens, Greece; 8Theoretical and Physical Chemistry Institute, National Hellenic Research Foundation, 11635 Athens, Greece

**Keywords:** quercetin, 2-hydroxyl-propyl-β-cyclodextrin, 2,6-methylated cyclodextrin, molecular interactions, NMR spectroscopy, molecular dynamics, absolute free energy calculation, FEP, fluorescence spectroscopy, Density Functional Theory (DFT)

## Abstract

Quercetin (QUE) is a well-known natural product that can exert beneficial properties on human health. However, due to its low solubility its bioavailability is limited. In the present study, we examine whether its formulation with two cyclodextrins (CDs) may enhance its pharmacological profile. Comparative interaction studies of quercetin with 2-hydroxyl-propyl-β-cyclodextrin (2HP-β-CD) and 2,6-methylated cyclodextrin (2,6Me-β-CD) were performed using NMR spectroscopy, DFT calculations, and in silico molecular dynamics (MD) simulations. Using T1 relaxation experiments and 2D DOSY it was illustrated that both cyclodextrin vehicles can host quercetin. Quantum mechanical calculations showed the formation of hydrogen bonds between QUE with 2HP-β-CD and 2,6Μe-β-CD. Six hydrogen bonds are formed ranging between 2 to 2.8 Å with 2HP-β-CD and four hydrogen bonds within 2.8 Å with 2,6Μe-β-CD. Calculations of absolute binding free energies show that quercetin binds favorably to both 2,6Me-β-CD and 2HP-β-CD. MM/GBSA results show equally favorable binding of quercetin in the two CDs. Fluorescence spectroscopy shows moderate binding of quercetin in 2HP-β-CD (520 M^−1^) and 2,6Me-β-CD (770 M^−1^). Thus, we propose that both formulations (2HP-β-CD:quercetin, 2,6Me-β-CD:quercetin) could be further explored and exploited as small molecule carriers in biological studies.

## 1. Introduction

Quercetin (QUE) (Figure 1) has shown potential health benefits in humans, and it can be used as a nutritional supplement in the pharmaceutical and food industries [1,2,3]. Like many flavonoids, QUE is best known for its antioxidant activity [4,5,6], suggesting that it could be used in preserving food quality by preventing the oxidative deterioration of lipids. Reported health benefits of QUE include cardiovascular protection, anti-ulcer effects, anti-allergic, antiviral, anti-inflammatory and antitumor activities [7,8,9,10]. Moreover, several studies report that QUE destabilizes and increases the clearance of abnormal proteins, which are key pathological marks of Alzheimer’s disease [11]. Therefore, QUE has emerged as a major bioactive ingredient with potential applications in many functional foods and pharmaceutical products. However, the poor water solubility of this phytochemical, limits significantly its bioavailability and absorption after oral administration. To overcome this problem, QUE has been formulated into novel lipid-based systems to improve its bioactivity and a therapeutic profile and nanotechnology approaches such as microparticles, nanostructured lipid carriers, nanoparticles, nanoemulsions, microemulsions, liposomes, phytosomes, niosomes and transferosomes have been applied [11,12,13,14,15,16,17]. Recent studies show that cellular penetration of QUE was enhanced by sterol containing solid lipid nanoparticles for targeting hepatocellular carcinoma cells [18]. It has also been reported that QUE-nanostructured lipid carriers (NLC) formulations can be a potential breakthrough for the treatment of breast cancer with minimal side-effects [19]. Laser diffraction size analysis of the aerosol and its excellent distribution and penetration capacity generated from QUE nanoemulsions shows its suitability for efficient pulmonary delivery for lung cancer treatment [20]. QUE lipid based formulations were explored for their anti-inflammatory activity, and it was concluded that these novel formulations exhibited better anti-inflammatory activity [21]. 

Cyclodextrins (CDs) are a group of cyclic oligosaccharides consisting of sugar monomers [α-D-glycopyranose], which are linked by α-[1,4]-glycosidic bonds. The natural CDs α, β and γ consist of 6, 7 and 8 glucose units, respectively. Apart from these naturally occurring CDs, many derivatives have been synthesized such as randomly methylated derivatives of β-CD [RM-β-CD], 2-hydroxypropylated β- and γ-CDs [HP-β-CD, HP-γ-CD], sulfobutylated-β-CDs [SBE-β-CD], branched CDs [glucosyl- and maltosyl-β-CDs], acetylated β- and γ-CDs and sulfated CDs. Their structures consist of a hydrophilic outer surface and a hydrophobic inner cavity [22,23]. The lipophilic cavity of CDs enables entrapment of hydrophobic molecules and the formation of host–guest complexes. Thus, CDs have been utilized in the food and pharmaceutical industries and have also been used for analytical purposes [24,25,26]. In the pharmaceutical industry, CDs have mainly been used as complexing agents to increase the aqueous solubility of active substances poorly soluble in water [27,28,29,30]. Various drugs have already been commercialized as complexes with CDs, including hydrocortisone, prostaglandin, nitroglycerin, itraconazol, and chloramphenicol [28]. CD complexes can be used to release biologically active compounds under specific conditions [31,32,33]. A recent study showed that the complexation with β-CD enhanced the therapeutic efficiency of praziquantel from 59% (for the plain drug) to 99% (for the β-CD complex) [34]. Moreover, studies with the essential oil of ocimum plant showed a low absorption giving a maximum % inhibition of edema equal to of 16%, while after its complexation with SBE-β-CD/HP-β-CD a higher absorption was observed achieving more than 50% inhibition of edema [35]. The curcumin-β-CD complex enhanced curcumin delivery and improved therapeutic efficacy compared to the free curcumin [36]. Furthermore, the encapsulation of phenoxodiol in β-CD revealed increased antiproliferative activity against cancer cells [37].

Numerous cases of natural product–CD complexes have been reported, such as for silibinin [38] and quercetin [39]. CDs and especially hydroxypropyl-CD have been widely used to protect drugs against conjugation and metabolic inactivation [40] as well as to enhance the aqueous solubility and the oral bioavailability of sparingly soluble drug molecules [41,42]. Encapsulation of natural product polyphenols into CDs has been performed to hinder the non-enzymatic and enzymatic dioxygen oxidation and to extend their stability over time [43]. The complexation of caffeic and rosmarinic acid with CDs could eliminate the aforementioned obstacles and additionally provide a shelter for the two molecules [44]. HP-β-CD is an isomer mixture with a different degree of hydroxyl-propylation [45]. HP-β-CD’s degree of substitution (DS) has a significant impact on the solubility and aggregation of the CD itself. Particularly, HP-β-CD reaches maximum solubility [46] while one or two DS results in the hindrance of the solubility potential [47]. The aggregation tends to decrease as the DS increases, while generally the solubility augments.

As a continuation of our previous studies, in order to analyze in depth both the 3D architecture as well as the thermodynamic properties and the interacting forces governing the formation of such host-guest systems, we synthesized two different encapsulation complexes of QUE with 2HP-β-CD and 2,6Me-β-CD (Figure 1). We also utilized an array of analytical and computational techniques including high resolution ^1^H NMR spectroscopy, T1 measurements, fluorescence spectroscopy, DFT calculations, molecular dynamics (MD) simulations and absolute binding free energy calculations to study QUE-CD interactions. High resolution ^1^H NMR spectroscopy utilizes the differences in chemical shifts between protons of QUE complexed with HP-β-CD and Me-β-CD and free forms. T1 measurements provide information for the components alone or in a complexed form. In the present work, we studied the QUE-CD interactions with two different types of computational methodologies, i.e., QM(DFT) and MD, adding physical insight on the interactions, and this approach makes the study “novel sufficient”. MD simulations, MM/GBSA calculations and rigorous alchemical non-equilibrium free energy calculations were performed to complement experimental results and provide the binding affinity of QUE upon complexing with the two CD systems. 

## 2. Results and Discussion

NMR analysis (carbon and proton NMR spectroscopy) is a powerful tool for the determination of the inclusion of a guest molecule inside CDs through the identification and comparison of the chemical shifts of the free guest molecule and CD with those of its complex. ^1^H-NMR analyses have been performed in the literature for the determination of structural changes during complexation of host-guest systems [48,49,50].

The ^1^H NMR spectra of the 2,6Me-β-CD and the lyophilized complex are shown in Figure 2. The chemical shifts of the peaks are shown in Table 1. Differences in chemical shifts between the two spectra are observed (Table 1) signifying the complexation between 2,6-Me-β-CD and QUE.

Chemical shifts were also calculated using the Self-Consistent Field Gauge-Independent Atomic Orbital (SCF GIAO) method and compared with the experimental results as shown in Table 2. The full spectra as calculated with the SCF GIAO method is shown in Appendix A. As it can be observed, experimental and theoretical calculations match closely.

Regarding the 2D NOESY NMR spectrum of the complex (full spectrum covering all regions is shown in Appendix A) depicts critical NOEs between the aromatic H2′*, H3′*, H6′* of QUE and the H2, H5 and H6 of 2,6Μe-β-CD. These results suggest that QUE enters entirely into the hydrophobic core of the 2,6Me-β-CD cavity.

The ^1^H NMR spectra of 2HP-β-CD and the lyophilized complex, in D_2_O at 25 °C are shown in Figure 3. The chemical shifts of the peaks are shown in Table 3. Chemical shifts were also calculated using the SCF GIAO method and compared with the experimental results as shown in Table 4, showing that experimental and theoretical calculations match closely. The full spectra as calculated with *the* SCF GIAO method is shown in Appendix A.

The 2D NOESY spectra of 2HP-β-CD-QUE (full spectrum shown in Appendix A) shows that the critical NOEs are between the aromatic H2′*, H3′*, H6′* of QUE and the H2″, H5″ and H6″ of 2HP-β-CD. The H8′, and H6′ of QUE present weaker NOEs compared to the respective protons of 2,6Me-β-CD. These results suggest that QUE enters entirely into the hydrophobic cavity of the 2HP-β-CD.

The experimental and theoretical values of QUE in the two complexes are shown in Table 5. Non-significant differences are observed in these values, although in some peaks the deviation is higher compared to the ones shown for the two CDs. This observation signifies that the theoretical calculation cannot simulate accurately the environment of all the protons emerged in the cyclodextrins core.

The complexation of QUE with CD induces upfield changes in the chemical shifts of the ^1^H-NMR spectra. In particular, H2′*, H3′* and H6′* show the highest chemical shift deviations suggesting intermolecular interactions between QUE and the two CDs.

The following tables (Table 6 and Table 7) and Appendix A include the relaxation times for the two complexes before and after the complexation.

The incorporation of quercetin in both CDs results in the decrease of T1 values implying strong host-guest interactions. The percentage of chemical shift changes indicates that QUE complexes stronger with 2,6Μe-β-CD compared to 2HP-β-CD.

### 2.1. 2D DOSY Experiment

Complex formation between QUE and both forms of cyclodextrin is evident from DOSY spectra, where the two components of the complex have the same apparent translational diffusion coefficient, Dt, equal to 1.9 × 10^−10^ m^2^ s^−1^ for both QUE:2,6Me-β-CD and QUE:2HP-β-CD 1:1 formulations. In DOSY spectra of both formulations, smaller signals are presented in the aromatic region of the ^1^H NMR spectrum, which have larger Dt values (around 3 × 10^−10^ m^2^ s^−1^). These findings indicate that a fraction of the QUE in both samples is unbound, and it is in a slow exchange equilibrium with the bound form (Figure 4 and Figure 5).

### 2.2. Conformational Analysis

The minimum energy conformations of QUE, 2HP-β-CD, 2,6Me-β-CD and the corresponding QUE-CD complexes have been calculated using DFT(B3LYP/6-31G(d,p) in water and are shown in Figure 6. QUE inserted into the cavity of CD, and its hydroxyl groups interact with the hydroxypropyl groups of 2HP-β-CD. 2,6Μe-β-CD encapsulates QUE in a similar fashion. It should be noted that to perform these calculations two initial starting configurations of QUE in CD were employed: (a) the resorcinol group interacting with the -OH rim of the CD and (b) the catechol group interacting with the -OH rim. Interestingly, the lowest energy encapsulation complex has the QUE molecule inserted in a different direction in the two CD molecules. The lowest energy structure is obtained when resorcinol group interacts with the -OH rim of the 2HP-β-CD and in the case of 2,6Μe-β-CD the lowest energy structure is obtained when the QUE catechol group interacts with the -OH and methoxy rim of 2,6Μe-β-CD. The remaining conformers, where the catechol of QUE interacts with the -OH rim (2HP-β-CD) and resorcinol of QUE interacts with the -OH and methoxy rim (2,6Μe-β-CD) are local energy minima lying energetically 3.77 kcal/mol and 3.36 kcal/mol above the global encapsulated minima for the two complexes, at the B3LYP/6-31g(d,p) level of theory (Figure 6). It should be noted that the less accurate semiempirical PM6 method (compared to the DFT methodology) also predicts the same encapsulation complexes as DFT does, but their energy differences are smaller, see Figure 6. 

The lowest energy structure for QUE forms two intramolecular hydrogen bonds, O…HO. However, when QUE is encapsulated in both CD molecules, these two intramolecular bonds break, and new intermolecular hydrogen bonds are formed between the OH of QUE and O and HO atoms of the CD molecules. Thus, the encapsulated conformer of QUE has not adopted the lowest energy structure but one which is 16.0 kcal/mol higher in energy. This conformer does not have any intramolecular bonds and the catechol group is twisted by 37.5 degrees with respect to the pyran group due to H … H repulsion. The hydrogen bonds between QUE with 2HP-β-CD (a) and 2,6Μe-β-CD (b) are shown in Figure 7. Six hydrogen bonds are formed that range from 2 to 2.8 Å in the case of 2HP-β-CD and four hydrogen bonds are formed with 2,6Μe-β-CD within 2.8 Å. It is important to note that binging energies of H...O interactions longer than 2.8 Å are almost zero [51]. The complexation energies and the deformation energies of the involved CD and QUE molecules are shown in Table 8.

All calculations were corrected via basis set superposition error (BSSE) [52]. We found that deformation energies, such as the energy difference between the structures of QUE and CDs within the complexes and the corresponding free conformer, are 3.89 and 20.15 kcal/mol for 2HP-β-CD (d1 of Figure 6) and 7.31 and 20.75 kcal/mol for 2,6Μe-β-CD (d2 of Fig11), respectively. These values show that the compounds are deformed to increase their interactions. The corrected BSSE binding energy is about 11 kcal/mol for 2HP-β-CD and e1 structure of 2,6Μe-β-CD, while it is 11.7 and 20.0 kcal/mol for the e1 and e2 structures, respectively. As a result, QUE can enter and exit reversibly from both CDs. However, the binding energy BEr with respect to the deformed structures of the molecules of the complex is large enough, namely, −33.0 kcal/mol and −11.5 (−24.2) kcal/mol, showing that the complexes can be isolated and as a result, they can be prepared for NMR experiments. This energy characterizes the structural changes of the complex with respect to the isolated molecules. 

### 2.3. MM/GBSA Estimates of the Free Energy of Binding Using MD Simulations

The binding free energy of QUE inside 2HP-β-CD and 2,6Me-β-CD for the first five representative structures of each system calculated from the all-atom MD simulations was assessed using the MM/GBSA method (Table 9). The first cluster representative structures for QUE/2HP-β-CD and QUE/2,6Me-β-CD complexes are depicted in Figure 8. The total binding free energy of QUE/2HP-β-CD is predicted to be −31.73 ± 3.21 kcal/mol, while the binding free energy of QUE/2,6Me-β-CD is −26.77 ± 2.37 kcal/mol. The energy decomposition into individual contributions shows that binding is driven mainly by significant van der Waals interactions that are similar for both complexes, while hydrophobic surface interactions also have a favorable contribution (−12.70 ± 1.81 and −6.26 ± 1.54 kcal/mol for 2HP-β-CD and 2,6Me-β-CD, respectively). Coulombic interactions between QUE and the two CD complexes contribute favorably to binding; however, the total electrostatic contribution is unfavorable due to solvent effects. The total binding free energy of both complexes, as calculated from MM/GBSA, suggests that QUE has favorable binding affinity for both CDs and thus CDs may be considered as effective carriers of QUE. However, a clear distinction of the binding preference of QUE cannot be made due to the error of the MM/GBSA method, which is on the order of several kcal/mol. Concluding, MM/GBSA predicts that both the 2HP-β-CD and 2,6Me-β-CD have a favorable binding affinity for QUE of similar magnitude, in agreement with the NMR experiments.

Comparing the BE of the DFT calculations with those of the MM/GBSA both methodologies are in very good agreement, i.e., −33.0 kcal/mol (DFT) and −31.73 ± 3.21 kcal/mol (MM/GBSA) for QUE/2HP-β-CD. In the case of the QUE/2,6Me-β-CD, the MM/GBSA calculated value of −26.77 ± 2.37 kcal/mol, see below, is in very good agreement with the second DFT calculated conformer of the encapsulated structures, i.e., −24.2 kcal/mol. 

### 2.4. Absolute Binding Free Energy Calculations

To further study the capability of CDs to act as potential carriers of QUE, we calculated the binding free energy of QUE inside 2HP-β-CD and 2,6Me-β-CD using MD simulations coupled with absolute free energy calculations and a non-equilibrium setup for faster convergence [53,54]. The work values obtained from the non-equilibrium TI simulations and the corresponding work distributions for both QUE/CD systems are depicted in Appendix A. Appendix A depict overlapping work distributions for the QUE solvation leg of the thermodynamic cycle for both systems, which is helpful to assess the sampling and convergence of the simulations using the overlap of forward and reverse work distributions. 

The calculated binding free energies (Table 10) show that both 2,6Me-β-CD (−5.09 ± 0.76 kcal/mol) and 2HP-β-CD (−1.40 ± 0.76 kcal/mol) have a favorable binding affinity for QUE with a ca. 3.5 kcal/mol preference for binding to 2,6Me-β-CD compared to 2HP-β-CD, indicating that both the 2HP-β-CD and 2,6Me-β-CD systems should be further studied as complexing agents to increase the aqueous solubility of QUE. These binding affinities are in agreement with the NMR results (Table 5) that showed that the percentage of chemical shift changes is more significant in the case of 2,6Μe-β-CD compared to 2HP-β-CD.

### 2.5. Fluorescence Spectroscopy 

To determine the interaction between QUE and CDs, fluorescence spectroscopy was performed. The fluorescence intensity of a small ligand can be altered upon the encapsulation in the cavity of a supramolecule such as CD [55,56] The spectroscopic behavior of the ligand was explored by adding increasing concentrations of either 2HP-β-CD or 2,6Me-β-CD into a specific concentration of QUE. Upon the gradual addition of the 2HP-β-CD, the fluorescent intensity of the formed physical mixture was enhanced by 1.84-fold while a 1.32-fold enhancement was recorded in the case of 2,6Me-β-CD (Figure 9). This particular enhancement of the fluorescence signal is usually observed, once a small molecule interacts with CD due to the changes occurring in the microenvironment of the small molecule upon encapsulation [57]. The binding constants of the complexes were derived from the Benesi−Hildebrand equation:(1)$$\frac{1}{\Delta F}=\frac{1}{\Delta F_{C}}+\frac{1}{K_{C}\Delta F_{C}\;{\left[CD\right]}_{0}}$$

The straight line of the double reciprocal plot confirms the 1:1 stoichiometry of QUE with CDs and the binding constants were calculated equal to 520 ± 32 M^−1^ and 771 ± 51 M^−1^ for 2HP-β-CD and 2,6Me-β-CD, respectively, indicating a moderate affinity between the two molecules (Figure 10).

## 3. Materials and Methods

### 3.1. Chemicals 

QUE (MW: 302.24 g/mol), 2,6Me-β-CD (MW: 1310 g/mol) and 2HP-β-CD (MW: 1460 g/mol) were purchased from Sigma-Aldrich (St. Louis, MO, USA), Fluka Chemika (Mexico City, Mexico US & Canada) and Ashland (Covington, KY, USA), respectively.

### 3.2. Preparation of the Complex

Lyophilized inclusion complex of QUE-Me-β-CD and QUE-HP-β-CD were prepared by freeze-drying aqueous solution of QUE-Me-β-CD and QUE-HP-β-CD, in molar ratio of 1:2, using the neutralization method, as described previously by Manta et al. [58]. Briefly, 2170 mg of 2,6Me-β-CD or 4800 mg of 2HP-β-CD were accurately weighed and transferred into a 600 mL glass vessel and suspended with 500 mL of purified water under magnetic stirring until complete dissolution of the cyclodextrin. Then, 500 mg of accurately weighed QUE were added and dissolved under continuous stirring and light protection (due to the photosensitivity of QUE), by adjusting the pH at approximately 9.0–9.5 with ammonium hydroxide solution 6% *v*/*v*. The volume of the obtained clear and colorless solution was fixed at 600 mL with purified water and was immediately frozen at −73 °C, and freeze-dried using Vacuum Freeze-Dryer, BK-FD10T, Biobase biodustry (Jinan, China) Co., Ltd.

### 3.3. High Resolution ^1^H NMR Spectroscopy

Samples were dissolved in 0.6 mL of D_2_O at a concentration of 10mM and transferred to 5 mm NMR tubes. 

DOSY spectra of QUE complexes were recorded on an Agilent Technologies VNMRS 8600 MHz NMR spectrometer with a 5 mm HCN cold probe. The DgcsteSL_cc sequence was used to record DOSY spectra with 65,536 points, 1 s relaxation delay, and 16 repetitions. Thirty-two gradient strengths between zero and 60 gauss/cm were used. All spectra were recorded at 25 °C. Chemical shifts are referenced with respect to the lock frequency and reported relative to TMS.

^1^H, NOESY and Τ1 experiments were performed on 400 MHz Bruker Avance using the spin-echo pulse sequence installed in the library of the NMR spectrometer. The variable delay list for T1 measurements contained 10 different values for τ that were applied between 90° and 180°. The calculation of T1 for each proton was done using the MestReNova program. The equation of three parameters used was:Β + F × exp(−x*G)(2)
where Β = magnetic induction field and F = spectral width and G = 1/T.

### 3.4. Conformational Analysis (Quantum Mechanics Calculations) 

Additionally, the interactions between the 2HP-β-CD and 2,6Μe-β-CD with QUE were calculated via DFT. At first, QUE was optimized using the B3LYP [59,60]/6-31g(d,p) [61] to find the lowest energy structure. Additionally, conformational analysis [52] was carried out for the 2HP-β-CD and 2,6Μe-β-CD to find the lowest energy structure using the B3LYP/6-31g(d,p). Subsequently, complexes with CD and QUE were optimized to find the lowest minima. All calculations were performed in water solvent employing the polarizable continuum model (PCM) [62]. Finally, the interaction energy between these two molecules was calculated and the binding energy was corrected using BSSE [63]. All the calculations and the visualization of the results were carried out via Gaussian 16 [64].

### 3.5. MD Simulations 

The crystal structure of β-CD was retrieved from the Cambridge Structural Database (CSD reference code: BUVSEQ02) [65] and was modified to 2-HP-β-CD and 2,6Me-β-CD using Schrodinger 2021.2. QUE was docked into the interior of both CDs using the GlideXP algorithm of the Schrodinger suite [66,67,68]. 

To model the systems the CHARMM General Force Field (CGenFF) was used to model QUE [69], and 2HP-β-CD and 2,6Me-β-CD were parameterized using the ADD force-field [70]. Each QUE/CD complex was solvated using the TIP3P water model [71] in a cubic box ensuring a minimum distance of 15 Å between each complex atom and the edge of the periodic box. The QUE/CD complexes were initially minimized for 50,000 steps using conjugate gradient. The resulting QUE/CD complexes were subjected to all-atom MD simulations using NAMD2.14 [72] with the following protocol. First the system was gradually heated to 300 K (NVT ensemble). Then, an equilibration protocol for 10 ns was run at 1.01325 bar and 300 K in the NPT ensemble. Finally, equilibrium MD simulations of each complex were conducted for 500 ns. Temperature control was maintained by a Langevin thermostat [73] (300 K) and pressure by a Nosé-Hoover Langevin barostat [74,75]. Particle-mesh Ewald method [76] was employed for long-range electrostatics interactions with a maximum grid spacing of 1 Å. Non-bonded interactions were calculated with a cutoff of 12 Å and a switching distance of 10 Å. A 2 fs time step was used and the SHAKE [77] algorithm was employed to constrain hydrogen atoms.

To obtain the five most representative structures of the CD complexes, we clustered the conformations of each complex during the last 400 ns of each simulation using the gromos [78] algorithm of the gmx_cluster routine (GROMACS 2020.6) [79]. Root mean square deviation (RMSD) on the non-hydrogen atoms of each CD was used as the clustering criterion and a cutoff value of 1.5 Å was chosen in order to obtain balanced cluster sizes. The central structures of the first five more populated clusters were picked in order to acquire the five most representative structures of each system.

MM/GBSA analysis was performed with Prime (Schrodinger Suite) [80,81] for the first five representative structures of each system. To calculate the binding free energy of QUE to both CDs the following equation was used:(3)$$\Delta G_{bind}=G_{complex}-G_{CD}-G_{QUE}$$
where *G_complex_*, *G_CD_* and *G_QUE_* are the free energies for the complex, the receptor (2HP-β-CD and 2,6Me-β-CD in this case) and the ligand (QUE), respectively. Δ*G_bind_* is a sum of coulombic interactions (Δ*G_Coulomb_*), van der Waals (Δ*G_vdW_*), generalised Born electrostatic solvation energy (Δ*G_SolvGB_*), hydrophobic surfaces interactions (Δ*G_Lipo_*), covalent energy (Δ*G_Covalent_*) and hydrogen-bonding corrections (Δ*G_Hbond_*). These parameters are calculated using the variable dielectric solvent model (VSGB2.0) [82] with the OPLS4 [83] force field as implemented in Prime (Schrodinger, Inc., New York, NY, USA).

The reported binding free energies reflect the mean of the calculations performed for the first five representative structures. Statistical uncertainties are provided as the standard deviation of these calculations.

### 3.6. Absolute Binding Free Energy Calculations

A well-established non-equilibrium free energy workflow [53,54] was used to calculate the absolute binding free energy of QUE with 2HP-β-CD and 2,6Me-β-CD. In this method, a double decoupling scheme [84,85] is employed to trace the alchemical path from QUE in solution to a fully interacting QUE/CD system as depicted in Figure 11.

First, QUE, in its physical unbound state (state I), is decoupled from the surrounding environment by annihilating its partial charges and its van der Waals parameters (state II) in order to calculate $\Delta G_{solv}^{elec+vdw}$. To keep the position and orientation of QUE close to that of a bound pose (state III), restraints, as defined by Boresch et al. [86], are introduced. The contribution of the added restraints $\left(\;\Delta G_{solv}^{restr}\right)$ is computed analytically with the protocol described by Boresch et al. [86]. By applying these restraints, state III can be assumed to be equivalent with a non-interacting molecule inside the CD cavity (state IV) since there are no interactions between QUE and CD. Finally, QUE interactions with the environment are turned back on (state V), providing the term $\Delta G_{prot}^{elec+vdw},$ and the restraints are removed (state VI) to obtain $\Delta G_{prot}^{restr}$. The free energies of QUE solvation ($\Delta G_{solv}^{elec+vdw}$) and CD/QUE coupling ($\Delta G_{prot}^{elec+vdw}+\Delta G_{prot}^{restr}$) are calculated separately by performing multiple non-equilibrium transitions between the QUE coupling and decoupling directions. For every transition, the Hamiltonian is linearly interpolated between the two end states using a λ increment of 4.5 × 10^−6^ per time step. The derivatives of the Hamiltonian with respect to λ are recorded to obtain the non-equilibrium work distributions. Finally, the maximum likelihood estimator [87] based on the Crooks fluctuation theorem [88] is used to relate the equilibrium free energy differences between the two end states with the resulting non-equilibrium work distributions.

Therefore, to calculate the absolute binding free energy of QUE to both CDs Equation (3) [89] can be used:(4)$$\Delta G_{bind}=\Delta G_{solv}^{elec+vdw}+\Delta G_{solv}^{restr}-\Delta G_{prot}^{elec+vdw}-\Delta G_{prot}^{restr}$$

The MD/TI simulations were carried out using GROMACS 2021.4. [90] Force-field parameters for QUE, 2HP-β-CD and 2,6Me-β-CD were identical to those used in the Section 3.5. The systems were solvated using TIP3P water in a cubic box with 15 Å of padding between solute and box edges.

Initially, each system was minimized for 10,000 steps, followed by 1 ns of NVT and NPT equilibration. Then, a further 20 ns NPT simulation was performed to derive the orientational Boresch restraints. The restraints for each QUE/CD complex were defined by applying the MDRestraintsGenerator algorithm [91] on the final 20 ns NPT simulation. Briefly, this algorithm works as follows: Firstly, the most stable atoms of QUE are chosen as anchor points for the restraints. Then, all available CD heavy atoms within an 8 Å cutoff of the QUE anchor atoms over the 20 ns simulation are selected, and a list of potential Boresch restraints is generated. The bond, angle and dihedral values throughout the NPT simulation for all identified restraints are recorded and the set of restraints with the lowest standard deviation across all values is chosen as the orientational restraint of choice. The frame closest to the mean bond, angle, and dihedral values of this set of restraints for each QUE/CD complex over the 20 ns simulation is then used as a starting point for the non-equilibrium thermodynamic integration (TI) simulations.

To equilibrate the systems, 1 ns NPT equilibration was performed using the Berendsen thermostat [92] with a time constant of 1 ps to maintain the pressure to 1 atm. Then, 20 ns NPT of equilibration followed, the pressure was kept at 1 atm using the Parrinello-Rahman barostat [93] with a time constant of 2 ps and a compressibility of 4.5 × 10^−5^ bar^−1^ and the temperature was maintained at 298 K through Langevin dynamics with a collision frequency of 2 ps^−1^. The particle-mesh Ewald method [79] was employed for long-range electrostatics interactions with a maximum grid spacing of 1 Å. Short range electrostatic and van der Waals interactions were calculated with a cutoff of 12 Å and a switching distance of 10 Å. A 2 fs time step was used and hydrogen atoms were constrained using the LINCS [94] algorithm.

To perform the non-equilibrium TI simulations, a short equilibration procedure similar in protocol to the aforementioned equilibration cycle was performed for both QUE solvation (state I → state II) and CD/QUE coupling (state V → state VI). This was followed by a 10 ns NPT production using the Parrinello-Rahman barostat [93] with a time constant of 2 ps and a compressibility of 4.5 × 10^−5^ bar^−1^. Then, 95 equidistant snapshots were extracted from each of the resulting trajectories after discarding the first 0.4 ns for equilibration. Finally, for every snapshot, 500 ps non-equilibrium transitions were performed in both coupling and decoupling directions.

Pmx [95] was used to compute the free energy estimates from the resulting work distributions in both directions using a maximum likelihood estimator [87] based on the Crooks fluctuation theorem [88]. Uncertainties were estimated via bootstrap. For each QUE/CD complex, the non-equilibrium workflow was repeated 4 times and the reported binding free energies ($\Delta G_{bind\;average})$ are the mean of the 4 replicates. Statistical uncertainties are provided by propagating the errors of the 4 replicates.

During the non-equilibrium TI simulations using GROMACS 2021.4, a non-bonded exclusion bug appeared as described in [96] and caused our simulations to stop. To solve this issue, we used the “couple-intramol = yes” flag, which causes the intra-molecular van der Waals and electrostatic interactions of QUE to be turned on and off during the simulations as suggested in [97] and [98]. The input files for these simulations are provided as SI information.

### 3.7. Fluorescence Spectroscopy Studies 

Fluorescence spectroscopy experiments were conducted to determine the binding constant of QUE with 2HP-β-CD and 2,6Me-β-CD. The experiments were performed in an Edinburg FS5 spectrofluorometer (Edinburgh Instruments Ltd., Livingston, UK). The excitation and emission slits were set at 5 nm and the emission spectra were recorded using a quartz (1 cm cuvette), at room temperature. A stock solution of QUE was prepared in DMSO/PBS buffer (10 mM, pH 7.4) [50:50 *v*/*v*%] at a concentration of 100 μΜ and kept in dark. The CD stock solutions were prepared in dH_2_O at a concentration of 6 mM. The final concentration of QUE in the cuvette was 25 μΜ for each measurement. Various volumes from the CDs stock solution were added each time (0, 0.1, 0.2, 0.3, 0.4, 0.5, 0.6, 0.7, 0.8, 0.9, 1.0, 2.0, 3.0 and 4.0 mM). The final volume of each sample was 3 mL, adjusted each time with the proper amount of dH_2_O. The samples were kept stirred and protected from light for 30 min, before measurement. The excitation wavelength of QUE was set at 375 nm. 

The binding constant between QUE and each CD was calculated based on the observed emission changes of the fluorescence spectrum upon the addition of different concentrations of 2HP-β-CD/2,6Me-β-CD. A titration curve at I550 was plotted by applying linear fitting. The binding constants were derived from the Benesi−Hildebrand equation:$$\frac{1}{\Delta F}=\frac{1}{\Delta F_{C}}+\frac{1}{K_{C}\Delta F_{C}\;{\left[CD\right]}_{0}}$$
where Δ*F* is the difference between the fluorescence intensities in the absence and presence of 2HP-β-CD/2,6MeβCD, *Kc* is the binding constant, Δ*F_C_* is the difference on intensity between free and complexed QUE at 1:1 molar ratio and [*CD*]_0_ is the concentration of 2HP-β-CD/2,6Me-β-CD.

## 4. Discussion and Conclusions

This study is a continuation of our previous study [57], where the interactions of 2,6-Me-CD with QUE were investigated and comparative studies were performed using only differential scanning calorimetry and solubility experiments. This additional study provides details on the molecular interactions between QUE and the two CDs through a variety of experimental and computational techniques. Using 2D NOESY experiments, the aromatic rings of QUE are clearly shown to be engulfed in the hydrophobic cavities of the two CDs. This finding was confirmed by MD simulations and DFT calculations. T1 relaxation experiments were very informative depicting the mobility of the protons in the free QUE and complexing form. The degree of their mobility reflects the interactions during the complexing in accordance with the 2D NOESY results. 

MD simulations were employed to investigate the binding and thermodynamic stability of the two CD/QUE encapsulation complexes. As the candidate drug can adopt various conformations inside the cavity, long equilibrium MD simulations were performed to obtain the five most representative structures of QUE inside the two CDs. These structures were then subjected to MM/GBSA analysis to estimate the binding free energy of QUE inside 2HP-β-CD and 2,6Me-β-CD. MM/GBSA is an approximate method to compute absolute binding affinities with small computational effort, which has been successfully used in previous studies to reproduce and analyze the experimental findings of inclusion CD complexes [99,100]. The MM/GBSA results show favorable binding of QUE to both CDs with −31.73 ± 3.21 kcal/mol for 2HP-β-CD and −26.77 ± 2.37 kcal/mol for 2,6Me-β-CD). However, this method involves several approximations including the lack of conformational entropy and the effect of water molecules, which limits its accuracy [101]. For this reason, rigorous absolute free energy calculations [102,103,104,105] using MD as the sampling technique were also performed here to evaluate the absolute binding affinities of the CD/QUE complexes. Free energy perturbation (FEP) calculations, derived from statistical mechanics, can accurately compute free energy differences associated with the binding of a small molecule to a host, albeit being computationally intensive. The robustness of absolute FEP calculations is evidenced from the number of prospective and retrospective applications [53,54,106,107,108,109,110,111], including the robust calculation of binding free energies of inclusion cyclodextrin complexes in the context of the Statistical Assessment of the Modeling of Proteins and Ligands (SAMPL7) challenge [112]. Here, our absolute binding free energy calculations confirmed favorable binding of QUE on CD with −1.40 ± 0.76 kcal/mol for 2HP-β-CD and −5.09 ± 0.76 kcal/mol for 2,6Me-β-CD. NMR spectroscopy results (Table 5) also indicate that QUE complexes stronger with 2,6Me-β-CD than 2HP-β-CD based on the percentage of chemical shift changes. However, FEP/MD calculations also come with limitations such as the type of restraints used to prevent the ligand drifting away from the host cavity [113]. In addition, the degree of overlap between forward and reverse work distributions, which is associated with the degree of sampling of the phase space along the followed alchemical pathway, could also be a limiting factor regarding the accuracy of the method [114]. This factor became evident also in our calculations as denoted by the non-overlapping work distributions for the QUE coupling leg of the thermodynamic cycle for both systems (Appendix A). The poor overlap of the distributions suggests inadequate sampling of the configurational sampling and could be tackled by increasing the number of transitions employed for the non-equilibrium protocol as well as increasing the simulation length of each transition. Certainly, despite the high computational cost and the current technical challenges, absolute FEP calculations is a promising strategy to computationally assess binding affinities of CD/QUE complexes.

Quantum mechanical calculations that were also employed herein showed the formation of hydrogen bonds between QUE with 2HP-β-CD and 2,6Μe-β-CD. In particular, six hydrogen bonds are formed that range between 2 to 2.8 Å in the case of 2HP-β-CD and four hydrogen bonds within 2.8 Å with 2,6Μe-β-CD. Moderate binding was revealed using fluorescence spectroscopy (520 M^−1^ for 2HP-β-CD and 770 M^−1^ for 2,6Me-β-CD).

The 2D DOSY experiment provided unequivocal evidence of the complexation of the quercetin with two CDs with the same apparent translational diffusion coefficient equal to 1.9 × 10^−10^ m^2^ s^−1^. In the DOSY spectra of both formulations, smaller signals are presented in the aromatic region of the ^1^H NMR spectrum, which appear with larger Dt values (around 3 × 10^−10^ m^2^ s^−1^). This result signifies that a fraction of the quercetin in both samples is unbound and is in a slow exchange equilibrium with the bound form. These studies explain at the molecular level our previous finding that the water-solubility of lyophilized QUE-Me-β-CD and QUE-HP-β-CD products were approximately 7–40-fold and 14–50-fold higher than pure QUE at pH 1.2–6. These results are encouraging for further ex vivo and in vivo evaluation for the nasal administration and nose-to brain delivery of QUE.

## Figures and Tables

**Figure 1 molecules-27-05490-f001:**
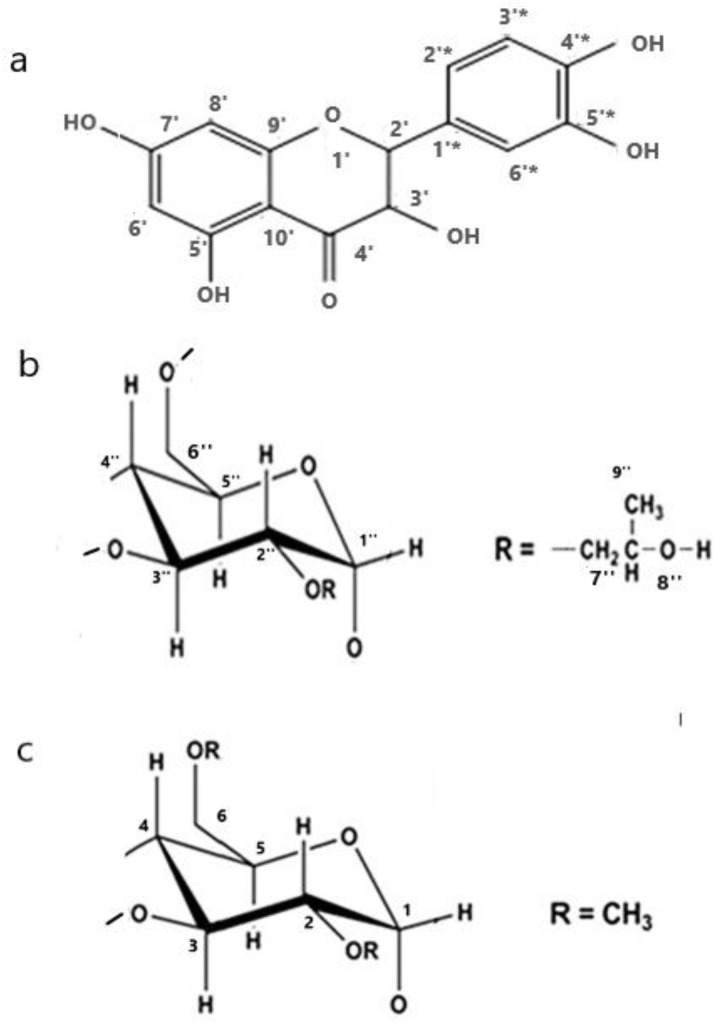
Chemical structure of (**a**) quercetin, (**b**) 2,6Μe-β-CD and (**c**) 2HP-β-CD.

**Figure 2 molecules-27-05490-f002:**
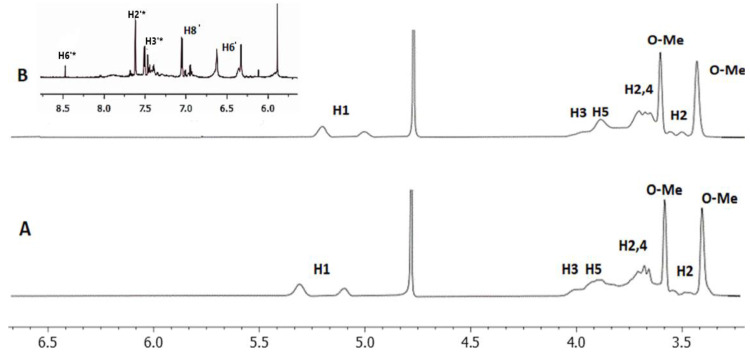
The ^1^H NMR of (**A**) 2,6Me-β-CD (**B**) complex of QUE with 2,6Me-β-CD. The spectra were obtained at 25 °C in D_2_O.

**Figure 3 molecules-27-05490-f003:**
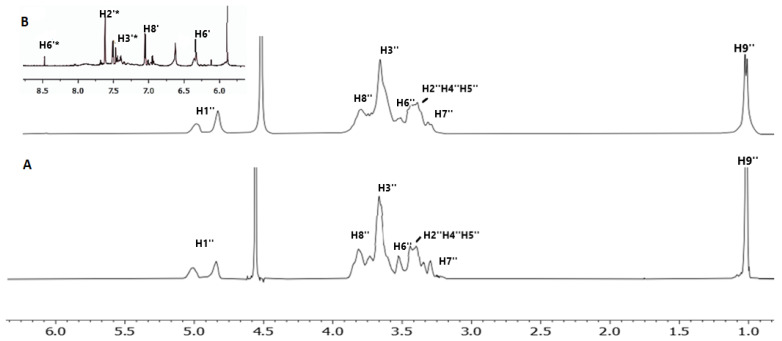
The ^1^H NMR of (**A**) 2HP-β-CD (**B**) complex of QUE with 2HP-β-CD. The spectra were obtained at 25 °C in D_2_O.

**Figure 4 molecules-27-05490-f004:**
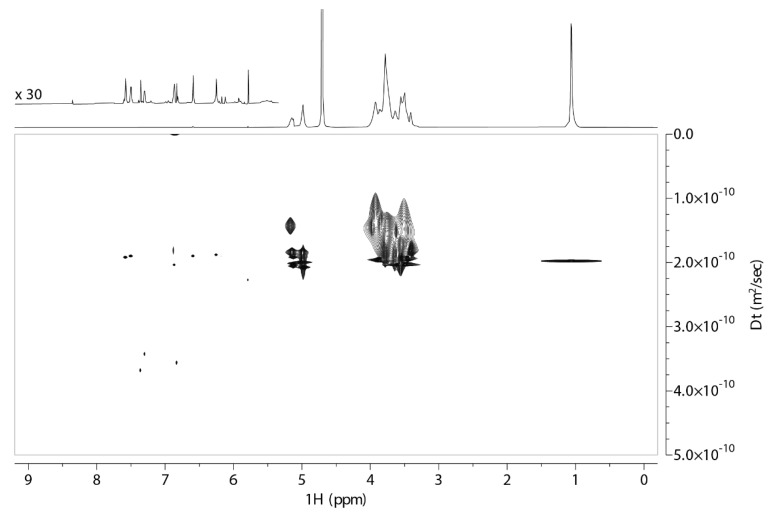
DOSY spectrum obtained of 2,6Me-β-CD-QUE (1:1) at 25 °C in D_2_O recorded on an 800 MHz NMR spectrometer. The corresponding ^1^H NMR spectrum is shown above, with an expansion of the aromatic region at 30-fold increased intensity. The recorded 2D DOSY experiment of the complex of QUE with 2,6Me-β-CD formation 1:1. The spectrum was obtained at 25 °C in D_2_O.

**Figure 5 molecules-27-05490-f005:**
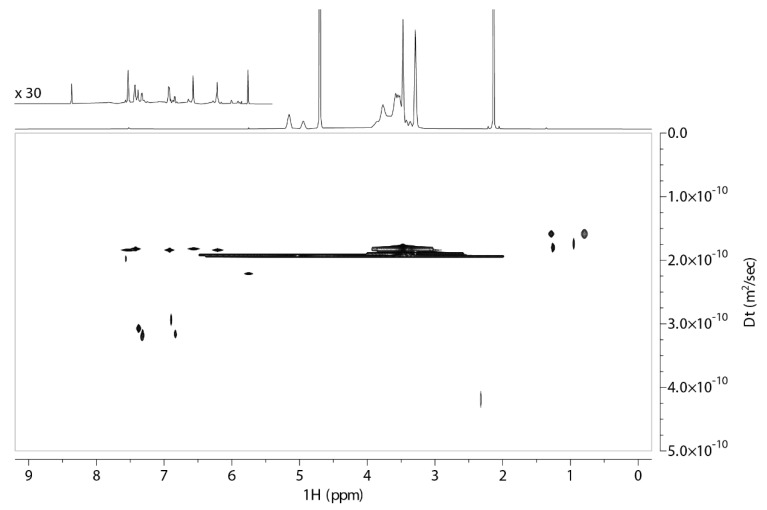
DOSY spectrum obtained of 2HP-β-CD-QUE (1:1) at 25 °C in D_2_O recorded on an 800 MHz NMR spectrometer. The corresponding ^1^H NMR spectrum is shown above, with an expansion of the aromatic region at 30-fold increased intensity.

**Figure 6 molecules-27-05490-f006:**
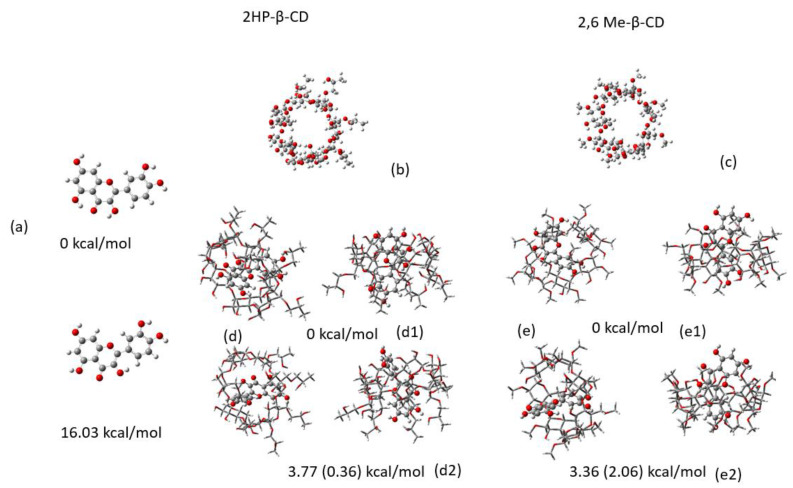
Calculated minimum energy structures: (**a**) QUE (above) and a local minimum of QUE (bottom), (**b**) 2HP-β-CD, (**c**) 2,6Μe-β-CD, (**d**) complex of QUE with 2HP-β-CD (**d1**) and complex of reversed QUE with 2HP-β-CD (**d2**) from two different points of view, and (**e**) complex of QUE with 2,6Μe-β-CD (**e1**) and complex of reversed QUE with 2,6Μe-β-CD (**e2**) from two different points of view. All calculations were performed in water solvent using B3LYP/6-31G(d,p). The energy differences between the conformations in complexes are shown for the B3LYP level of theory; energy differences calculated using PM6 are shown in parentheses.

**Figure 7 molecules-27-05490-f007:**
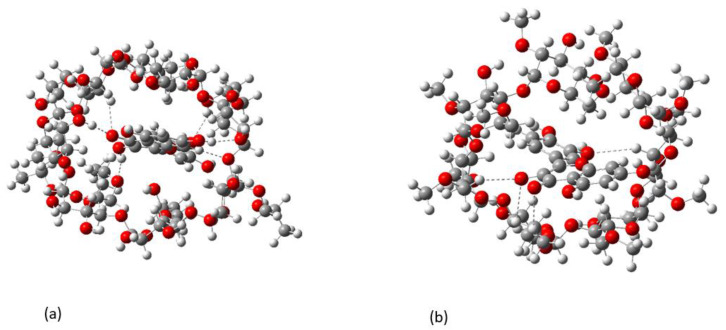
Conformations of the QUE complexed with 2HP-β-CD (**a**) and 2,6Μe-β-CD (**b**). Hydrogen bonds are shown with dashed lines.

**Figure 8 molecules-27-05490-f008:**
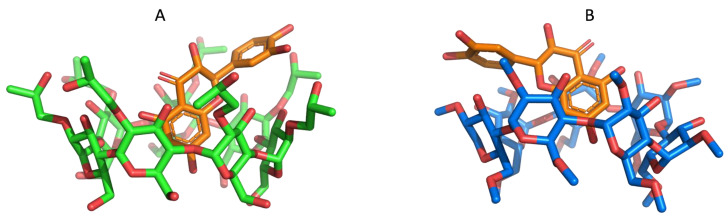
Molecular models of QUE (represented as orange sticks) inside the two cyclodextrin systems calculated as the first cluster representatives from MD simulations. (**A**) 2,6Me-β-CD (green sticks) and (**B**) 2HP-β-CD (blue sticks). Oxygen atoms are represented in red.

**Figure 9 molecules-27-05490-f009:**
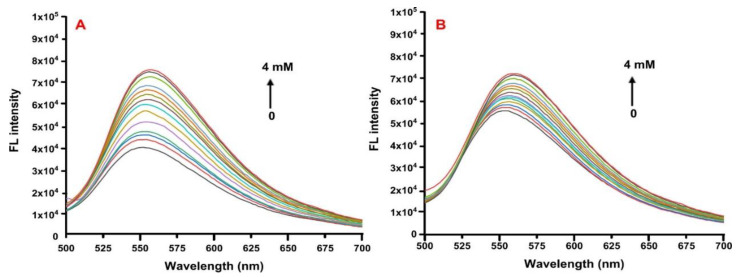
Fluorescence spectra of QUE (25 μM) at pH 7.4 (DMSO/PBS buffer, 10 mM) after titrating with various concentrations of CDs (0, 0.1, 0.2, 0.3, 0.4, 0.5, 0.6, 0.7, 0.8, 0.9, 1.0, 2.0, 3.0 and 4.0) (**A**) 2HP-β-CD and (**B**) 2,6-Me-β-CD.

**Figure 10 molecules-27-05490-f010:**
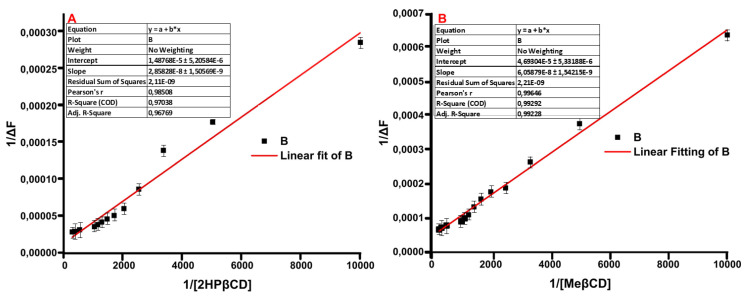
Double-reciprocal plots of the QUE titration with various concentrations of CDs (0, 0.1, 0.2, 0.3, 0.4, 0.5, 0.6, 0.7, 0.8, 0.9, 1.0, 2.0, 3.0 and 4.0) (**A**) 2HP-β-CD and (**B**) 2,6Me-β-CD.

**Figure 11 molecules-27-05490-f011:**
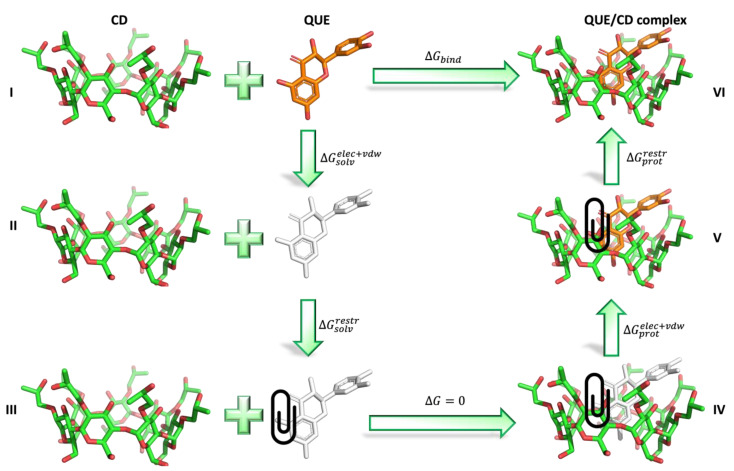
Thermodynamic cycle for absolute binding free energy calculations. QUE in solution (state I) is depicted as orange sticks and CD as green sticks. The presence of restraints is indicated with a paperclip. QUE depicted with white sticks means that the molecule is decoupled from the solvent.

**Table 1 molecules-27-05490-t001:** ^1^H NMR chemical shifts for 2,6Me-β-CD and the lyophilized complex, in D_2_O at 25 °C.

Protons of 2.6 Μe-β-CD	Chemical Shifts	Chemical Shifts of Complex	Δδ (ppm)	Multiplicity
H1	5.35−5.09	5.29−5.03	0.02	singlet
H3	4.01	4.00	0.01	singlet
H5	3.94	3.90	0.04	doublet
H6	3.91	3.88	0.03	singlet
H2	3.72	3.68	0.04	singlet
H4	3.68	3.67	0.01	doublet
H6-O-Me	3.59	3.59	0	singlet
H2	3.49	3.44	0.05	singlet
H2-O-Me	3.41	3.42	0.01	singlet

**Table 2 molecules-27-05490-t002:** ^1^H NMR chemical shifts using the SCF GIAO method and experimental results for the 2,6Me-β-CD in the complex.

Protons of 2,6Μe-β-CD	Chemical Shifts of Complex Experimental Results	Chemical Shifts of Complex SCF GIAO Method
H1	5.29−5.03	5.30−5.07
H3	4.00	4.01
H5	3.90	3.90
H6	3.88	3.89
H2	3.68	3.68
H4	3.67	3.67
H6-O-Me	3.59	3.55
2	3.44	3.43
H2-O-Me	3.42	3.40

**Table 3 molecules-27-05490-t003:** ^1^H NMR chemical shifts for 2HP-β-CD and the lyophilized complex, in D_2_O at 25 °C.

Protons of 2HP-β-CD	Chemical Shifts	Δδ (ppm)	Multiplicity
H1″	5.03−4.92	0.01	singlet
H8″	3.87	0	singlet
H3″	3.75	0	singlet
H6″	3.58	0.04	singlet
H2″-H4″-H5″	3.47	0.04	multiple peaks
H7″	3.37	0.01	singlet
H9″	1.01	0	singlet

**Table 4 molecules-27-05490-t004:** ^1^H NMR chemical shifts using the SCF GIAO method and experimental results for the HP-β-CD in the complex.

Protons of 2HP-β-CD	Chemical Shifts of Complex Experimental Results	Chemical Shifts of Complex SCF GIAO Method
H1″	5.03−4.92	5.08−4.90
H8″	3.87	3.84
H3″	3.75	3.75
H6″	3.54	3.57
H2″-H4″-H5″	3.43	3.40
H7″	3.36	3.36
H9″	1.01	1.00

**Table 5 molecules-27-05490-t005:** ^1^H NMR chemical shifts for QUE and its complexes with 2,6 Me-β-CD and 2HP-β-CD in D_2_O at 25° and chemical shifts obtained using the SCF GIAO method.

Protons of QUE	Chemical Shifts	Complex with 2,6 Me-β-CD Experimental	Complex with Me-β-CD Using SCF GIAO Method	Complex with 2HΡ-β-CD Experimental	Complex with 2HΡ-β-CD Using SCF GIAO Method	Multiplicity
H6′	7.87	7.65	7.73	7.70	7.70	singlet
H2′*	7.65	7.57	7.50	7.60	7.30	singlet
H3′*	7.15	7.05	7.35	7.02	7.00	doublet
H8′	6.65	6.59	6.27	6.69	6.44	doublet
H6′*	5.90–5.93	6.06–6.09	5.99–6.01	5.86–5.89	5.80–5.90	broad singlet

**Table 6 molecules-27-05490-t006:** T1 measurement for 2HΡ-CD before and after complexation.

Peak Name	Τ1 2HΡ-β-CD	T1 2HP-β-CD with QUE	Percentage Change
H1″	1.40 s−1.54 s	1.39 s−1.52 s	2%
H8″	1.72 s	1.73 s	1%
H3″	1.23 s	1.23 s	0%
H6″	1.44 s	1.28 s	16%
H2″-H4″-H5″	1.42 s	1.20 s	22%
H7″	1.55 s	1.55 s	0%
H9″	1.19 s	1.20 s	1%

**Table 7 molecules-27-05490-t007:** T1 measurement for 2,6Μe-β-CD before and after complexation.

Peak Name	Τ1 2,6Μe-β-CD	Τ1 2,6Μe-β-CD with QUE	Percentage Change
H1	1.36 s−1.4 s	1.35 s−1.4 s	1%
H3	1.49 s	1.51 s	2%
H5	1.41 s	1.18 s	23%
H6	1.34 s	1.12 s	22%
H2	1.36 s	1.12 s	24%
H4	1.35 s	1.28 s	7%
H6-O-Me	1.76 s	1.79 s	3%
H2	1.51 s	1.39 s	12%
H2-O-Me	1.64 s	1.62 s	2%

**Table 8 molecules-27-05490-t008:** Binding energies, BE, corrected values for BSSE, BEBSSE, and binding energies with respect to the deformed structures of the molecules of the complex, BEr (kcal/mol), deformation energies of the QUE and cyclodextrin, DefL and DefCD., at B3LYP/6-31G(d,p). All energies are expressed in kcal/mol.

	2HP-β-CD (d_1_) ^a^	2HP-β-CD (d_2_) ^a^	2,6Μe-β-CD (e_1_) ^a^	2,6Μe-β-CD (e_2_) ^a^
Deformation_QUE	3.89 (19.91) ^b^	3.42 (19.44) ^b^	0.59 (16.62) ^b^	7.31 (23.34) ^b^
Def_CD	20.15	18.99	11.75	20.75
BE	−8.91 (7.12) ^b^	−10.56 (5.46) ^b^	0.83 (16.85) ^b^	3.85 (19.87) ^b^
BE(BSSE)	10.73 (26.76) ^b^	10.90 (26.93) ^b^	11.68 (27.71) ^b^	20.03 (36.06) ^b^
BE_raw	−32.95	−32.97	−11.52	−24.21
BE(BSSE)_fcp	−13.31	−11.50	−0.66	−8.03

^a^ See Figure 1; ^b^ with respect to the global minimum conformer of QUE.

**Table 9 molecules-27-05490-t009:** Binding free energy analysis for the QUE/2HP-β-CD and QUE/2,6Me-β-CD complexes as obtained by the MM/GBSA calculations.

Energy Component (kcal/mol)	2HP-β-CD	2,6Me-β-CD
MM/GBSA ΔG_vdW_	−27.35 ± 1.10	−27.90 ± 0.92
MM/GBSA ΔG_Coulomb_	−2.08 ± 1.36	−5.37 ± 1.90
MM/GBSA ΔG_SolvGB_	+9.13 ± 0.85	+9.99 ± 1.65
MM/GBSA ΔG_Lipo_	−12.70 ± 1.81	−6.26 ± 1.54
MM/GBSA ΔG_Hbond_	−0.48 ± 0.16	−0.54 ± 0.13
MM/GBSA ΔG_Covalent_	+1.69 ± 0.79	+1.63 ± 0.73
MM/GBSA ΔG_bind_	−31.73 ± 3.21	−26.77 ± 2.37

**Table 10 molecules-27-05490-t010:** Absolute free energies of binding (kcal/mol) for the QUE/2HP-β-CD and QUE/2,6Me-β-CD complexes calculated by non-equilibrium TI simulations for four different simulation replicas.

Systems under Study	Replica1	Replica2	Replica3	Replica4	$\Delta G_{bind}\;average$
2HP-β-CD	−1.45 ± 0.46	−1.39 ± 0.28	−1.55 ± 0.40	−1.21 ± 0.36	−1.40 ± 0.76
2,6Me-β-CD	−5.21 ± 0.37	−5.14 ± 0.28	−4.53 ± 0.45	−5.47 ± 0.39	−5.09 ± 0.76

## Data Availability

Input files for MD simulations andabsolute free energy calculations using GROMACS 2021.4 are deposited in this link: https://doi.org/10.5281/zenodo.6967423 (accessed on 9 June 2022).

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
