# Peer review of "Comparative Interaction Studies of Quercetin with 2-Hydroxyl-propyl-β-cyclodextrin and 2,6-Methylated-β-cyclodextrin"

_molecules, 2022, doi:10.3390/molecules27175490_

Round 1
Reviewer 1 Report
The manuscript describes research on the incorporation of quercetin into cyclodextrins. The cyclodextrin formulation of various drugs has been extensively studied, so the topic is somewhat worn out. However, the authors used both experimental methods and simulations to gain more detailed insight into guest-host interactions. The research is well planned AND conducted. However, the quality of data presentation and interpretation of the results should be improved. Below are some comments that the authors should consider for improvements.
Figure 1, the chemical structures of cyclodextrins are wrong. The compounds shown in this figure are not 2-hydroxyl-propyl-β-cyclodextrin (2HP-β-CD) and 2,6-methylated cyclodextrin (2,6Me-β-CD).
Table 1 does not shown the chemical shifts for QUE.
It is not clear, what is shown in Figure 2. The authors are suggested to include the chemical structure of the compounds with proton assignments in this figure. Also, the proton labels for different compounds should be different. Similarly, Figure 3.
Page 9, “shown in Figure 10” it is rather Figure 6.
How was the number of hydrogen bonds calculated?
Page 12, the Benesi−Hildebrand equation should be shown on this page, since it is mentioned here for the first time.
The quality of the data depicted in Figure 10 is poor. Did the authors preform repetitions of the experiments?
The conclusions should be rewritten to present the most important outcomes from the results.
Author Response
Question #1
The manuscript describes research on the incorporation of quercetin into cyclodextrins. The cyclodextrin formulation of various drugs has been extensively studied, so the topic is somewhat worn out. However, the authors used both experimental methods and simulations to gain more detailed insight into guest-host interactions. The research is well planned AND conducted. However, the quality of data presentation and interpretation of the results should be improved. Below are some comments that the authors should consider for improvements.
Figure 1, the chemical structures of cyclodextrins are wrong. The compounds shown in this figure are not 2-hydroxyl-propyl-β-cyclodextrin (2HP-β-CD) and 2,6-methylated cyclodextrin (2,6Me-β-CD).
Answer #1
We would like to thank the reviewer for this point that has been properly revised.
Question #2
Table 1 does not shown the chemical shifts for QUE.
Answer #2
We would like to thank the reviewer for this point that has been properly revised.
Question #3
It is not clear, what is shown in Figure 2. The authors are suggested to include the chemical structure of the compounds with proton assignments in this figure. Also, the proton labels for different compounds should be different. Similarly, Figure 3.
Answer #3
We thank reviewer for the comment. All the protons of quercetin and cyclodextrins are shown with different labeling to avoid confusion. The legends appear to be clear.
Question #4
Page 9, “shown in Figure 10” it is rather Figure 6.
Answer #4
Reviewer is right and Figure 10 was converted to Figure 6.
Question #5
How was the number of hydrogen bonds calculated?
Answer #5
In 2.2 section, the conformation analyses of the QUE, CD and QUE in CD have been carried out via DFT methodology, i.e., all atoms have calculated quantum mechanically. Thus, we have characterized as hydrogen bonds the H...O interactions having H...O distances shorter than 2.8 Å, (See p.10, lines:262-264). Note, that H...O interactions longer than 2.8 Å are almost zero.
Question #6
Page 12, the Benesi−Hildebrand equation should be shown on this page, since it is mentioned here for the first time.
Answer #6
The Benesi-Hildebrand equation was inserted as has been suggested by the reviewer.
Question #7
The quality of the data depicted in Figure 10 is poor. Did the authors preform repetitions of the experiments?
Answer #7
The experiments were conducted in triplicates so the error bars were incorporated in the relevant diagram and the resolution was also increased.
Question #8
The conclusions should be rewritten to present the most important outcomes from the results.
Answer #8
The conclusions have been rewritten to present the most important outcomes from the results as it is suggested by the reviewer.
Reviewer 2 Report
Comments to Author:
The manuscript "Comparative interaction studies of quercetin with 2-hydroxyl-propyl-β-cycodextrin and 2,6-methylated-β-cyclodextrin" by Vakali and coworkers describes an investigation of the comparative interaction of quercetin with 2-hydroxyl-propyl-β-cyclodextrin (2HP-β-CD) and 2,6-methylated cyclodextrin (2,6Me-β-CD) using NMR spectroscopy and in silico Molecular Dynamics (MD) simulations. As a general comment, I consider this study as well done and characterized by results of great interest. In my opinion, this manuscript deserves to be published in Molecules, and I suggest minor revision:
Comment
The 2,6-methylated cyclodextrin (2,6Me-β-CD) is not an adequate name for the compound. Double check and change throughout the text. Check Figure 1 with the compounds, the names do not match
Author Response
Comments to Author:
The manuscript "Comparative interaction studies of quercetin with 2-hydroxyl-propyl-β-cycodextrin and 2,6-methylated-β-cyclodextrin" by Vakali and coworkers describes an investigation of the comparative interaction of quercetin with 2-hydroxyl-propyl-β-cyclodextrin (2HP-β-CD) and 2,6-methylated cyclodextrin (2,6Me-β-CD) using NMR spectroscopy and in silico Molecular Dynamics (MD) simulations. As a general comment, I consider this study as well done and characterized by results of great interest. In my opinion, this manuscript deserves to be published in Molecules, and I suggest minor revision:
Comment
The 2,6-methylated cyclodextrin (2,6Me-β-CD) is not an adequate name for the compound. Double check and change throughout the text. Check Figure 1 with the compounds, the names do not match
Answer
The reviewer is absolutely right. We thank him very much for his comments and for his positive view on the quality of the work.
Round 2
Reviewer 1 Report
The manuscript has been improved and can be acceped for publication.
Author Response
Comment:
In the present work, we studied the QUE-CD interactions with two different types of computational methodologies, i.e., QM(DFT) and MD, adding physical insight on the interactions, and this approach makes the study "novel sufficient". MD simulations, MM/GBSA calculations and rigorous alchemical non-equilibrium free energy calculations were performed to complement experimental results and provide the binding affinity of QUE upon complexing with the two CD systems. This has been incoporated in the introduction of the manuscript.
In addition we performed linguistic improvements to the whole manuscitpt